# Genetic Analysis of Torque Teno Canis Virus Identified in Republic of Korea

**DOI:** 10.3390/vetsci9120693

**Published:** 2022-12-13

**Authors:** Da-Yoon Kim, Hee-Seop Ahn, Sang-Hoon Han, Hyeon-Jeong Go, Dong-Hwi Kim, Jae-Hyeong Kim, Joong-Bok Lee, Seung-Yong Park, Chang-Seon Song, Sang-Won Lee, In-Soo Choi

**Affiliations:** 1Department of Infectious Diseases, College of Veterinary Medicine, Konkuk University, 120 Neungdong-ro, Seoul 05029, Republic of Korea; 2Konkuk University Zoonotic Diseases Research Center, College of Veterinary Medicine, Konkuk University, 120 Neungdong-ro, Seoul 05029, Republic of Korea; 3Konkuk University Center for Animal Blood Medical Science, College of Veterinary Medicine, Konkuk University, 120 Neungdong-ro, Seoul 05029, Republic of Korea

**Keywords:** dog, Torque teno canis virus, incidence, recombination, Republic of Korea

## Abstract

**Simple Summary:**

Torque teno viruses are small, ubiquitous viruses with highly diverse genomes and a wide host range. However, their incidence in dogs remains unknown in Republic of Korea. In this study, we detected Torque teno canis virus and performed whole-genome sequencing using canine fecal samples. This is the first study to determine the incidence of Torque teno canis virus in Republic of Korea and report a new recombinant virus.

**Abstract:**

Torque teno canis virus (TTCaV) is an approximately 2.8 kb circular single-stranded DNA virus known to cause infections in dogs. However, its incidence in Republic of Korea remains unknown. In this study, 135 dog fecal samples were collected to determine TTCaV infection status in Republic of Korea. Based on polymerase chain reaction (PCR) analysis, 13 of 135 (9.6%) dogs tested positive for TTCaV. Three full-length genome sequences (GenBank IDs: MZ503910, MZ503911, and MZ503912) were obtained from the positive specimens. Phylogenetic tree construction and sequence identity, similarity plot, and recombination analyses were performed using these three full-length genomic sequences. Among the three full-length genomes, MZ503912 was determined to be a recombinant virus based on analysis with the reference TTCaV strains. This novel virus strain might have been generated by recombination between TTCaV strain KX827768 discovered in China and MZ503910 discovered in Republic of Korea. This is the first report to determine the incidence, genetic variation, and recombination of TTCaV in dogs in Republic of Korea. Further studies are needed to elucidate TTCaV pathogenesis in dogs.

## 1. Introduction

Torque teno virus (TTV) was first identified in 1997 in a Japanese patient with post-transfusion hepatitis of unknown etiology [1]. Human TTVs are nonenveloped, negative-sense, single-stranded, circular DNA viruses with a genome length of 3.6–3.9 kb [2]. TTVs belong to the family *Anelloviridae* [3]. The seroprevalence of human TTV in Brazil during 1997–1998 was 41.3% and the prevalence of TTV infection was reported to increase gradually with age [4]. Human TTVs exhibit an extremely wide sequence diversity, and five genogroups and several genotypes have been identified so far [5,6,7,8,9]. Human TTVs are transmitted by parenteral, trans-placental, breast milk, saliva, and fecal–oral routes [10,11,12,13].

TTVs have also been detected in non-human primates and other mammals. The complete nucleotide sequences of species-specific TTV that infect non-human primates, such as chimpanzees (*Pan troglodytes*), Japanese macaques (*Macaca fuscata*), cotton-top tamarins (*Saguinus oedipus*), and douroucouli (*Aotes trivirgatus*), have been reported [14,15]. Furthermore, full-length nucleotide sequences of TTV-infecting mammalian species, such as Tupaia (Tbc-TTV14), pigs (Sd-TTV31), dogs (Cf-TTV10), and cats (Fc-TTV4), have been determined [16,17].

Torque teno canis virus (TTCaV) was first identified in 2002 [16]. TTCaV harbors an approximately 2.8 kb circular single-stranded DNA as its genome. The TTCaV genome contains three open-reading frames (ORF1, ORF2, and ORF3) and an untranslated region (UTR) with a GC content of up to 90% [18]. TTCaV are classified under the genus *Thetatorquevirus* of the family *Anelloviridae* [3]. The family *Anelloviridae* also includes viruses that infect animals, such as pigs (Torque teno sus virus), cats (Torque teno felis virus), and tupaias (Torque teno tupaia virus) [3,12]. Human anelloviruses, especially TTVs, have been implicated in various diseases, such as hepatic and pulmonary diseases, hematological and autoimmune disorders, idiopathic inflammatory myopathy, and cancers [12,19]. However, the pathogenesis of TTV infections is not fully understood because of the lack of a cell culture system for viral propagation or suitable small animal models [20].

Several cases of TTCaV infection in dogs have been reported worldwide using serum and fecal samples [16,18,21,22]. However, to the best of our knowledge, there are no scientific reports on TTCaV incidence in dogs in Republic of Korea. Therefore, in this study, we investigated the incidence and genetic characteristics of TTCaV in Republic of Korea.

## 2. Materials and Methods

### 2.1. Sample Collection

A total of 135 canine fecal samples were collected in Republic of Korea. In 2019, 29 and 32 fecal samples were collected in Anseong and Seoul, respectively. In 2021, 74 fecal samples were collected in Yangpyeong. Yangpyeong and Anseong samples were collected from animal shelters, while Seoul samples were collected from the Veterinary Medical Teaching Hospital, Konkuk University. All the samples were suspended in 10-times volume (*w/v*) of PBS and centrifuged at 3000× *g* for 15 min. The fecal supernatants were stored at −80 °C until further analysis.

### 2.2. Viral DNA Extraction

Viral DNA was extracted from fecal supernatants using the Patho Gene-spin™ DNA/RNA Extraction Kit (iNtRON Biotechnology, Seongnam, Republic of Korea) according to the manufacturer’s instructions. The isolated DNA was eluted in 40 μL distilled water and stored at −20 °C for further experiments.

### 2.3. TTCaV Detection

We designed primer set I (TTCaV-F1 and TTCaV-R1, Table 1) for TTCaV detection by aligning full-length genome sequences of TTCaV. The representative sequences of TTCaV were obtained from GenBank (https://www.ncbi.nlm.nih.gov/genbank/, accessed on 19 January 2021) (AB076002, KX827767, MK050988, MK050987, KX827769, KX827768, GU951508, KX827770, KX827771, and KX377522). PCR was performed to detect TTCaV and each PCR mixture contained 5 μL template DNA, 1 μL primers (20 μM each), 5 U FastStart High Fidelity Enzyme Blend (Roche Diagnostic, Mannheim, Germany), 5 μL reaction buffer, 1 μL PCR grade nucleotide mix, 1 μL DMSO, and 35 μL distilled water to make up the volume to 50 μL. The amplification was initiated by preheating the mixture for 3 min at 95 °C, followed by 34 cycles of 30 s at 95 °C, 30 s at 45.4 °C, and 1 min at 72 °C; and a final 5 min extension at 72 °C. The amplicon was resolved on a 2% agarose gel and purified using the MEGAquick-spin Plus Total Fragment DNA Purification Kit (iNtRON Biotechnology, Seongnam, Republic of Korea). The purified amplicon was sequenced by Cosmo Genetech (Seoul, Republic of Korea).

### 2.4. PCR Amplification of Entire TTCaV Genome

TTCaV-positive samples were used to amplify the ORF1 region using primer set II (ORF1-F and ORF1-R, Table 1). The PCR mixture comprised 5 μL template DNA, 1 μL primers (20 μM each), 5 U TaKaRa LA Taq (Takara Korea Biomedical, Seoul, Republic of Korea), 25 μL of 2× GC buffer I, 8 μL dNTP mixture (2.5 mM each), and 9 μL distilled water to make up the volume to 50 μL. PCR amplification was initiated by preheating the reaction mixture for 3 min at 95 °C, followed by 34 cycles of 30 s at 95 °C, 30 s at 49 °C, and 2 min at 72 °C; and a final 5 min extension at 72 °C.

Primer set III (TTCaV-F2 and TTCaV-R2, Table 1) was used to amplify the remaining TTCaV-containing GC-rich regions. The total volume and components of the PCR were the same as those used for the ORF1 amplification. PCR amplification was initiated by preheating the reaction mixture for 3 min at 95 °C, followed by 34 cycles of 30 s at 95 °C, 30 s at 54.5 °C, and 2 min at 72 °C; and a final 5 min extension at 72 °C.

The amplicons were purified using a MEGAquick-spin Plus Total Fragment DNA Purification Kit (iNtRON Biotechnology, Seongnam, Republic of Korea) and cloned into the RBC T&A cloning vector system (RBC Bioscience, Taipei, Taiwan) according to the manufacturer’s instructions. The clone containing the amplified DNA was selected and sequenced by Cosmo Genetech (Seoul, Republic of Korea).

### 2.5. Phylogenetic Tree and Sequence Identity Analysis

The full sequences of TTCaV were aligned using the ClustalW multiple alignment tool of the Bio-Edit software (Ibis Biosciences, Carlsbad, CA, USA), and used for phylogenetic tree generation and sequence identity analysis. Phylogenetic trees were generated by the maximum likelihood method with 1000 bootstrap replicates using the MEGA-X software (Pennsylvania State University, PA, USA). Sequence identity analysis was performed using the Bio-Edit software (Ibis Biosciences, Carlsbad, CA, USA). The representative sequences of TTV and TTCaV were obtained from GenBank (GenBank number is shown in Figure 1).

### 2.6. Similarity Plot and Recombination Analysis

The full genome alignment of TTCaV was used for similarity plot and recombination analyses. Similarity plot analysis was performed using SimPlot software (Johns Hopkins University, Baltimore, MD, USA) and the Kimura 2-parameter model; the Japanese TTCaV prototype (AB076002) was used as a query. The following seven methods in the recombination detection program 4 (University of Cape Town, Cape Town, South Africa) were used to screen for potential recombinations and breakpoints: RDP, GENECONV, BootScan, Maxchi, Chimaera, SiScan, and 3Seq. The highest acceptable *p* value cut-off was 0.05. The representative sequences of TTCaV were obtained from GenBank (GenBank number is shown in Figure 2).

## 3. Results

### 3.1. TTCaV Incidence in Dogs

TTCaV was detected using primer set I, which amplified an 847 bp (169–1015 bp, Table 1) amplicon in 13 of 135 fecal samples, indicating an incidence of 9.6% (Table 2). Specifically, the incidence of TTCaV in the three provinces, Yangpyeong, Anseong, and Seoul in Republic of Korea, was 5%, 24%, and 6%, respectively (Table 2).

### 3.2. Complete Genomic Analysis of TTCaV

In the study, we used primer set 1 for the detection of TTCaV in canine fecal samples, which amplified 847 bp of TTCaV. We used the PCR products for DNA sequencing and determined the detection rate of TTCaV. We selected 600 bp from the 847 bp by removing unclear sequences including the primer regions, and the beginnings and ends of the sequences. Then, we analyzed the phylogenetic tree drawn by 600 bp (Appendix A). We selected the three isolates, Anseong 4, Anseong 13, and Anseong 18, which were located in different branches in the phylogenetic tree. We finally obtained three full genomic sequences from the three isolates and their sequences were deposited in GenBank (MZ503910 (2793 bp), MZ503911 (2795 bp), and MZ503912 (2791 bp)).

Phylogenetic analysis using complete TTV genomic sequences showed that these three full-length TTCaV genomes belonged to *Thetatorqueviruses* with representative strains Cf-TTV10 (AB076002), LDL (GU951508), and Sh-TTV203 (HM855265) (Figure 1).

The sequence identity analysis of the three complete genomes using the Japanese prototype TTCaV (AB076002) revealed homology ranging between 88.4 to 97.5%. The three sequences showed 88.2–96.9% nucleotide identities. The three TTCaV strains harbored all the three ORFs as expected. The ORF1 of the three strains encoded 576, 577, and 575 amino acids, whereas ORF2 and ORF3 in the three strains encoded 101 and 243 amino acids, respectively (Table 3).

### 3.3. SimPlot and Recombination Analysis of TTCaV

SimPlot analysis was performed to compare the nucleotide similarity of the three full TTCaV sequences, MZ503910, MZ503911, and MZ503912, with the prototype strain AB076002. Among the three sequences, MZ503912, especially the central sequence of ORF1, showed lower similarity than the other two strains (Figure 2). These data indicate that it may be a recombinant virus.

Next, recombination analysis was performed with representative strains of TTCaV and TTCaVs identified in this study. The MZ503912 was determined to be a recombinant variant. When the recombination events were analyzed using recombination detection programs, all seven methods indicated a recombination event with *p* < 0.05 (Appendix A). Recombination likely occurred between the MZ503910 strain isolated from Republic of Korea and the KX827768 strain isolated from China as a minor and major parent, respectively (Figure 3); of the 2791 bp MZ503912 genome, the region spanning 47–992 bp was derived from the minor parent MZ503910 strain, and the regions spanning 1–46 and 993–2791 bp were derived from the major parent KX827768 strain (Figure 3).

## 4. Discussion

In the present study, TTCaV incidence in Korean dogs was 9.6%. Earlier, TTCaV incidence was reported as 7% and 38% in canine serum samples from China and Japan, respectively [16,18]. TTCaV can be detected not only in sera, but also in feces; the TTCaV detection rate was 13% when fecal samples were used from dogs younger than 1-year-old in China [22]. In this study, we used only fecal samples collected from three different regions. The incidence rates of TTCaV in the Yangpyeong, Anseong, and Seoul samples were 5%, 24%, and 6%, respectively. A similar study performed in Turkey showed that TTCaV incidence, determined using canine fecal samples collected from a shelter, was 32% [21]. Therefore, the higher TTCaV incidence in the Anseong region could be attributed to the larger population of dogs in the shelter.

The human TTV genome consists of a UTR and a coding region comprising ORF1, 2, 3, and 4. Most mutations translating into amino acid substitutions occur in the hypervariable region of ORF1 in the human TTV genome [12]. SimPlot analysis was conducted to compare the sequence identity of ORF1, ORF2, and ORF3 of the three full genomic sequences of TTCaV strains, MZ503910, MZ503911, and MZ503912, using the Japanese prototype TTCaV AB076002. The central part of ORF1 of the MZ503912 strain showed a relatively lower sequence identity than those of MZ503910 and MZ503911. However, a hypervariable region of ORF1 in the TTCaV genome has not yet been reported. Subsequent analysis of the full genome sequence of MZ503912 with other full viral genomes revealed that the MZ503912 variant was generated by recombination between the MZ503910 strain isolated in this study and the KX827768 strain isolated in China. Human TTVs show wide genetic diversity and recombination among variants occurs frequently [19]. Homologous recombination within and among genotypes has been reported for TTVs infecting humans [24,25]. However, recombination in TTCaVs has not yet been reported. Thus, this is the first study to demonstrate TTCaV recombination. Further studies are needed to elucidate whether TTV evolves by recombination, resulting in its high incidence and variability.

## 5. Conclusions

In this study, we determined TTCaV incidence in Republic of Korea. We identified three complete genomic sequences of TTCaV and identified a new recombinant strain. Further studies are needed to determine the mechanism of recombination and its role in TTCaV evolution. Our results will facilitate further studies focusing on TTCaV pathogenesis in dogs.

## Figures and Tables

**Figure 1 vetsci-09-00693-f001:**
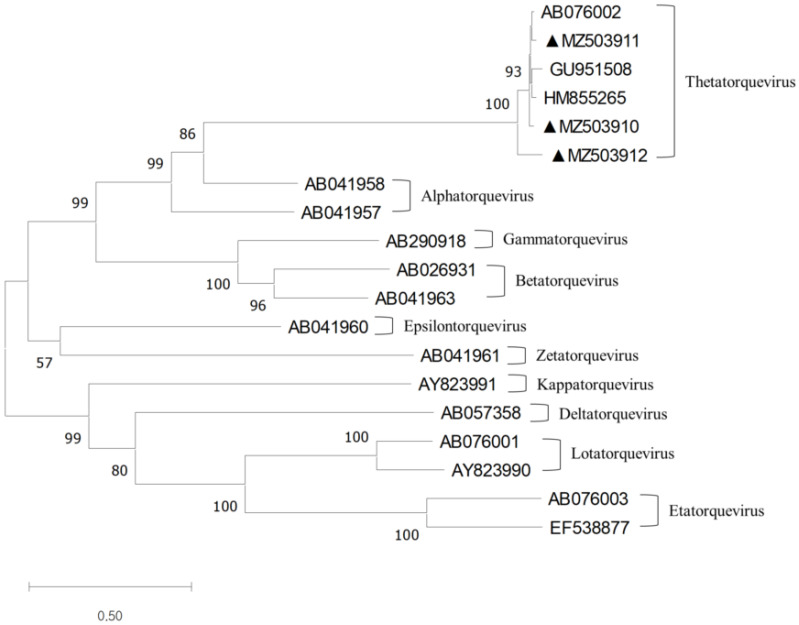
Phylogenetic tree constructed using the full-length TTCaV genomes and the reference TTV strain genomes. The maximum likelihood method and 1000 bootstraps were used. The branches with a bootstrap value greater than 70 were considered strong bootstrap support (*p* < 0.05) [23]. The three TTCaV isolates identified in this study are marked with closed triangles and were grouped into *Thetatorquevirus*.

**Figure 2 vetsci-09-00693-f002:**
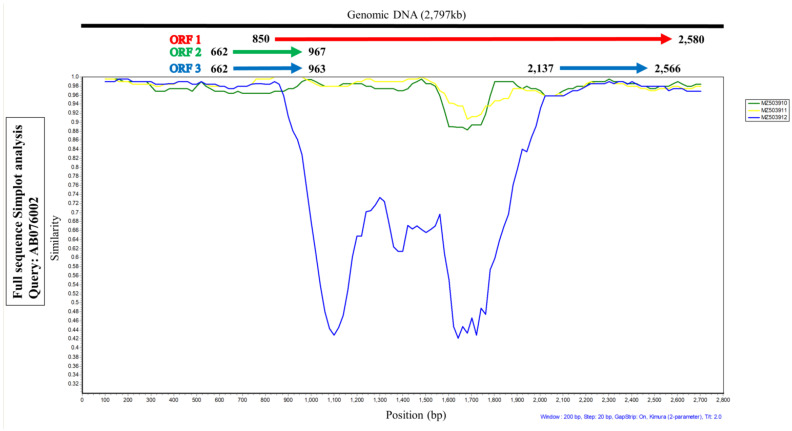
Similarity plot analysis using the three full-length genomes of TTCaV isolated in this study and the Japanese prototype Cf-TTV10 (query). ORF1 region of the MZ503912 strain showed lower similarities than the other two TTCaV strains.

**Figure 3 vetsci-09-00693-f003:**
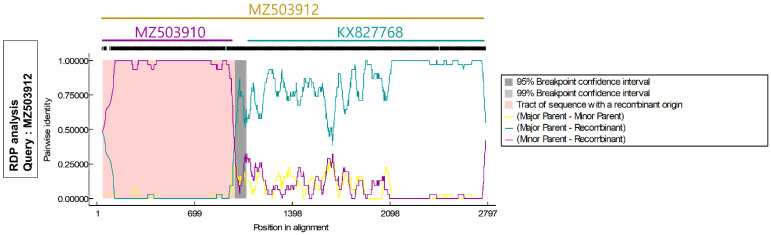
Recombination analysis using full-length TTCaV genome. MZ503912 was considered a recombinant of MZ503910 strain (minor parent) isolated from Republic of Korea and KX827768 strain (major parent) isolated from China.

**Table 1 vetsci-09-00693-t001:** Primer sets used for the detection of partial and full TTCaV genomic sequences.

Primer Sets	Primer Names	Sequences (5′→3′)	Positions *	Amplicon Length (bp)
I	TTCaV-F1	CGCCATCTTGGATTGGAAATC	169–189 993–1015	847
	TTCaV-R1	TAGAAATGTATTGTCTTTTGGTG		
II	ORF1-F	AAGCAGCACTGTAGCTGGAG	770–789 2613–2634	1865
	ORF1-R	CTTACGTCACAAAACAAGATGG		
III	TTCaV-F2	ATGGTGGCCCATTACCAACCCCTAC	1911–1935 247–273	1160
	TTCaV-R2	TATTCCGATGTCCGATTTGCATAATCG		

* Primer positions are indicated based on the genome of the prototype Japanese TTCaV (AB076002).

**Table 2 vetsci-09-00693-t002:** TTCaV incidence in different regions in Republic of Korea.

Region	Number of Fecal Samples	Number of TTCaV-Positive Fecal Samples	Incidence (%)
Yangpyeong	74	4	5
Anseong	29	7	24
Seoul	32	2	6
Total	135	13	9.6

**Table 3 vetsci-09-00693-t003:** Comparison of ORF1, 2, and 3 of three strains with TTCaV prototype (AB076002).

Accession Number	Genome Length (nt)	ORF1 aa	ORF2 aa	ORF3 aa
AB076002	2797	576	101	243
MZ503910	2793	576	101	243
MZ503911	2795	577	101	243
MZ503912	2791	575	101	243

## Data Availability

All the data generated or analyzed in this study are included in this published article and its Appendix A.

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
