# Peer review of "Genetic Analysis of Torque Teno Canis Virus Identified in Republic of Korea"

_vetsci, 2022, doi:10.3390/vetsci9120693_

Round 1

Reviewer 1 Report

The manuscript by Kim et al entitled “Genetic analysis of Torque teno canis virus identified in Korea” aims to report the incidence of Torque teno canis virus (TTCaV) in Korea for the first time. The authors analyzed 135 fecal samples from 3 regions in South Korea and identified 9.6% of samples positive for TTCaV. Subsequently, they performed full-genome sequencing of three of these samples. Analysis of these sequences determined that they all clustered within the Thetatorquevirus genus. Interestingly, similarity and recombination analysis suggested that one of these sequences may represent an ORF1 recombinant with a strain reported in China. Overall, this is a simple and straightforward study that adds some new information (in the form of sequence data) to the currently limited literature on TTCaV. It is sufficient for a case report but could have been strengthened with additional sequence analysis. However, there are several comments that the authors should address:

1.     The authors refer as “prevalence” when they have actually looked at samples collected within a period of time spanning from 2019-2021. Thus, the term “incidence” would be more suitable. Please modify this throughout the manuscript including abstract

2.     How did the authors determine that 135 samples would be an adequate number for determination of incidence? Under sample collection, it would be important to indicate how many samples per year are represented in order to have a better perspective of temporal context and determine whether any given year might be biased based on specimens collected. 

3.     The study is limited to 3 full length sequences. What was the justification/rationale to pick these three? Did they belong to different geographical regions and from which year? The study would have been strengthened by sequencing all the samples, which would have strengthened the analysis as well and made it suitable for a full-length research manuscript.

4.     All tables are missing in the manuscript and, thus, cannot be reviewed. 

5.     L77-78: something is missing in this sentence, please correct and provide GenBank number for the sequence used. 

6.     What is the target for primer set I? 

7.     L97-98: Does this sample set amplify ORF2 through the end of the genome? Please make sure the tables contain this information for each amplicon generated. 

8.     L127: Which is the ORF amplified by primer set I? 

9.     The legend for figure 3 should be expanded to better understand the methodology and terminology (e.g. minor vs major parent). Also, GenBank number is incomplete in the legend. 

10.  L177: sera instead of “serums”

11.  L186: the authors cite a hypervariable region of ORF1. Which is this region and have the authors looked at this site? Is it the site involved in the recombination event? How variable is this region among the sequenced samples both at nucleotide and amino acid levels? 

Author Response

Point-by-point Response to Reviewer 1 Comments  

Point 1.     The authors refer as “prevalence” when they have actually looked at samples collected within a period of time spanning from 2019-2021. Thus, the term “incidence” would be more suitable. Please modify this throughout the manuscript including abstract.

Response 1- We modified the term “prevalence” to “incidence” in our manuscript.

Point 2.     How did the authors determine that 135 samples would be an adequate number for determination of incidence? Under sample collection, it would be important to indicate how many samples per year are represented in order to have a better perspective of temporal context and determine whether any given year might be biased based on specimens collected. 

Response 2- In our previous studies, we determined the detection rates of hepatitis E virus in rabbits and pigs with their fecal samples. In those studies, we collected 126 rabbit samples (Virus Genes, 2018, 54: 587-590. Evidence of hepatitis E virus infection in specific pathogen-free rabbits in Korea. Han et al.) and 148 pig samples (J Vet Sci, 2018, 19: 309-312. Detection of hepatitis E virus genotypes 3 and 4 in pig farms in Korea). Based on those experience, we thought the 135 canine fecal samples might be enough of determining the detection of TTCaV. We indicated the years, regions, and places of sample collection in the Materials and Methods (lines 70-74) as follows. We collected a total of 135 fecal samples, 74 in Yangpyeong, 29 in Anseong, and 32 in Seoul. Anseong and Seoul samples were collected in 2019, and Yangpyeong samples were collected in 2021. So we used 61 samples collected at 2019, and 74 samples collected at 2021. Since there are no samples collected in 2020, we deleted the period “2019 to 2021” in the text.

Point 3.     The study is limited to 3 full length sequences. What was the justification/rationale to pick these three? Did they belong to different geographical regions and from which year? The study would have been strengthened by sequencing all the samples, which would have strengthened the analysis as well and made it suitable for a full-length research manuscript.

Response 3- In the study we used primer set 1 for detection of TTCaV in canine fecal samples, which amplified 847 bp of TTCaV. We used the PCR products for DNA sequencing and determined the detection rate of TTCaV. We selected 600 bp from the 847 bp by removing unclear sequences including the primer regions and the beginnings and ends of the sequences. And then we analyzed the phylogenetic tree drawn by 600 bp (Supplementary figure 1). We selected the three isolates Anseong 4, Anseong 13, and Anseong 18 which located in different branches in the phylogenetic tree. We finally obtained three full genomic sequences from the three isolates as shown in Fig 1. All three full length sequences are from Anseong district in Korea in 2021.

Point 4.     All tables are missing in the manuscript and, thus, cannot be reviewed. 

Response 4- We put the table in text.

Point 5.     L77-78: something is missing in this sentence, please correct and provide GenBank number for the sequence used. 

Response 5- We listed the reference sequences in the text.

Point 6.     What is the target for primer set I? 

Response 6- We provided the sequences, positions, and product length of the primer sets in Table 1 (line 97).

Point 7.     L97-98: Does this sample set amplify ORF2 through the end of the genome? Please make sure the tables contain this information for each amplicon generated. 

Response 7- We used three primer sets to amplify the full sequence of TTCaV. We listed the sequences, positions, and product length of the primer sets used in this case report in Table 1 (line 97). Primer set 1 amplifies ORF2, and primer set 3 amplifies ORF3 and a part of ORF1.

Point 8.     L127: Which is the ORF amplified by primer set I? 

Response 8- Primer set 1 is used for detection of TTCaV and contains both non-coding region and coding region. This primer set can amplify ORF2. Please refer Table 1 (line 97).

Point 9.     The legend for figure 3 should be expanded to better understand the methodology and terminology (e.g. minor vs major parent). Also, GenBank number is incomplete in the legend. 

Response 9- We provided more information in figure 3 and the legend of it.

Point 10.  L177: sera instead of “serums”

Response 10- We modified “serums” to “sera”.

Point 11.  L186: the authors cite a hypervariable region of ORF1. Which is this region and have the authors looked at this site? Is it the site involved in the recombination event? How variable is this region among the sequenced samples both at nucleotide and amino acid levels? 

Response 11- Several hypervariable regions of ORF1 are known in human TTV. However, the hypervariable region in TTCaV has not been determined. In this case report, we found that in MZ503912, where recombination occurred, the central part of ORF1 showed relatively lower sequence identity. So TTCaV is also expected to have a hypervariable region in ORF1.

Reviewer 2 Report

The authors analyzed the prevalence of TTCaV in Korean cities using fecal samples. They could obtain full-length viral sequences and deposit them into the database. I don't have significant concerns, but some information will be informative to the readers.

1. How does the virus transmission to the dogs? It will be a piece of good information in the introduction.

2. The authors collected 135 fecal samples from 3 cities. Are they collected from the specific animal facility in each city?  If they collected the samples from a specific facility, there may be a bias for sampling because it is possible of clustering of infection.

3. Did the dog, which showed positive in fecal, have any symptoms?
And, is there any viral preference for dog species? It would be good information for the readers.

Author Response

Point-by-point Response to Reviewer 2 Comments

Point 1. How does the virus transmission to the dogs? It will be a piece of good information in the introduction.

Response 1- The main route of transmission of TTV is believed to be by blood. However, because of the global prevalence and ubiquity of TTV virus, other routes are also considered. TTV have been detected in saliva, water, food, and feces. So oral and fecal-oral transmission is also considered to be the transmission route of TTV. The transmission route of TTCaV is not known yet. So far, TTCaV have been detected from sera and feces of dogs. We describe the TTV transmission mode in text (lines 43-45 and 64-65). We also provided the related reference in the introduction section [10-13,16,18,21,22].

Point 2. The authors collected 135 fecal samples from 3 cities. Are they collected from the specific animal facility in each city?  If they collected the samples from a specific facility, there may be a bias for sampling because it is possible of clustering of infection.

Response 2- A total of 135 fecal samples were collected from three different regions. Yangpyeong and Anseong samples were collected from animal shelters, and Seoul samples were collected from the Veterinary Medical Teaching Hospital, Konkuk University. As you concerned, there could be a clustering of infection in samples collected from animal shelters. However, other studies detected TTCaV in animal shelters. Therefore, we also conducted our study by referring the previously published studies (line 200-201).

Point 3. Did the dog, which showed positive in fecal, have any symptoms?
And, is there any viral preference for dog species? It would be good information for the readers.

Response 3- There is partial information of dogs which visited the Small Animal Hospital, Konkuk University, Seoul, Korea. Their symptoms were not restricted to gastrointestinal signs. Therefore, we did not provide their symptoms in our manuscript. In addition, we don’t have information about the samples obtained from the animal shelters. Also, clear correlation between certain animal symptoms and TTCaV is not known yet. Therefore, we did not describe each dog’s symptoms or clinical signs in this case report. We just tried to get the TTCaV incidence rates from fecal samples.

Round 2

Reviewer 1 Report

The authors have adequately addressed the reviewers' comments.